# Yield-Related Traits of 20 Spring Camelina Genotypes Grown in a Multi-Environment Study in Serbia

**Boris Kuzmanović** [1], **Sofija Petrović** [1], **Nevena Nagl** [2], **Velimir Mladenov** [1], **Nada Grahovac** [2], **Federica Zanetti** [3], **Christina Eynck** [4], **Johann Vollmann** [5] **and Ana Marjanović Jeromela** [2,*]

1   Faculty of Agriculture, University of Novi Sad, 21000 Novi Sad, Serbia; kuzmanovic.boris@gmail.com (B.K.); sonjap@polj.uns.ac.rs (S.P.); velimir.mladenov@polj.edu.rs (V.M.)
2   Institute of Field and Vegetable Crops, Maksima Gorkog 30, 21000 Novi Sad, Serbia; nevena.nagl@ifvcns.ns.ac.rs (N.N.); nada.grahovac@ifvcns.ns.ac.rs (N.G.)
3   Department of Agriculture and Food Sciences, Alma Mater Studiorum-Università di Bologna, 40127 Bologna, Italy; federica.zanetti5@unibo.it
4   Saskatoon Research and Development Centre, Agriculture and Agri-Food Canada, Saskatoon, SK S7N 0X2, Canada; christina.eynck@canada.ca
5   Institute of Plant Breeding, University of Natural Resources and Life Sciences—BOKU, 1180 Vienna, Austria; johann.vollmann@boku.ac.at
*   Correspondence: ana.jeromela@ifvcns.ns.ac.rs; Tel.: +381-648-205-739

**Abstract:** *Camelina sativa* (L.) Crantz is one of the oldest oilseed crops in Europe. Over the last twenty years, it has reemerged as a very promising alternative oilseed crop. Camelina has broad environmental adaptability, a wide range of resistances to pests and diseases, low-input requirements, and multiple industrial and feed applications exist for its seed oil and meal. In a multi-environment study conducted in Serbia, seven yield-related traits, including plant height (PH), height to the first branch (HFB), number of lateral branches (NLB), number of seed capsules per plant (NSCP), number of seeds per plant (NSP), mass of seeds per plant (MSP), and the total percentage of oil in the seed (TPOS), were analyzed in 20 spring camelina accessions. The combination of two years, two locations, and two sowing dates (autumn and spring) resulted in eight different environments across which the performance of the accessions was evaluated. The aims of the study were (a) to provide a phenotypic characterization and performance evaluation of the camelina accessions, (b) to identify correlations between the selected traits, and (c) to determine the effect of environmental factors on the traits. Environments contributed to the largest proportion in the total variance, explaining approximately 90% of the variance for all traits, except for NLB (70.96%) and TPOS (42.56%). The additive main effects and multiplicative interaction model (AMMI) showed that the weather conditions, and seeding dates were the most influential environmental factor. Location had a minor to moderate effect on the investigated traits. Lines CK3X-7 and Maksimir had the highest seed yields, and CK2X–9 and CJ11X–43 had the highest seed oil contents. All four lines had good adaptability and yield stability, making them the most suitable candidates for cultivation in the environmental conditions of Serbia in southeastern Europe. The present results reveal a potential for developing higher-yielding camelina cultivars with increased seed oil content and improved adaptability to various environmental conditions.

**Keywords:** *Camelina sativa*; environment; AMMI; yield traits; correlation

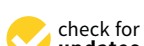



## 1. Introduction

*Camelina sativa* (L.) Crantz, also known as "false flax" or "gold of pleasure", is a self-pollinated, annual oilseed that belongs to the *Brassicaceae* family. Today's reemerging interest in camelina exists not just because it is a multipurpose oilseed crop, but also because it possesses a wide range of positive attributes that make it more adaptable to diverse environmental conditions than most other oilseed crops [1]. Camelina has a long history

of cultivation; it is assumed that it originated most likely from southeastern Europe and southwestern Asia, although the exact region of its origin is still unknown [2]. Archeological findings dating back as far as the Neolithic suggest that camelina represents one of the oldest oilseed crops that have been cultivated in Western Europe and Scandinavia. It was widely grown throughout the northern hemisphere, mainly in Europe and Russia, until the first half of the 20th century, when it was replaced by higher-yielding crops, such as rapeseed and sunflower [3].

Currently, camelina is emerging as an alternative oilseed crop worldwide, because it possesses a high seed oil content that is rich in antioxidants and essential fatty acids, particularly the OMEGA-3 fatty acid linolenic acid, and because it has a number of environmental stability traits such as tolerance to drought and cold that make this plant more suitable for production on low-quality soils and under harsh environmental conditions. Additionally, camelina is more resistant to most diseases and pests that generally attack oilseed crops [4–6]. The only limitations for camelina in terms of growing environments are heavy clay soils in combination with heavy precipitation and severe drought during sensitive growth stages, such as germination and flowering [7,8].

When comparing camelina and rapeseed (*Brassica napus* L. var. *oleifera*), as its closest crop relative, camelina can outperform rapeseed in environments with marginal growing conditions. Even though seed yield and oil content are strongly influenced by environmental conditions and genotype × environment interactions, the combination of camelina's adaptability traits and its low fertilizer and pesticide demand make it an attractive option for less productive lands and areas with low amounts of precipitation [1,9,10].

Camelina is a short-season crop that has a vegetation cycle of 85–100 days [11]. Today, a lot of research is being conducted to explore the potential uses of camelina as an oilseed crop. One of the potential applications of camelina is as an oilseed crop that can be used for biofuel and bio-lubricant production (high-quality fuel that has reduced greenhouse gas emissions by 40%) [12]. Other potential uses for camelina oil include the production of paints, lubricants, inks, soaps, cosmetics, and as a plastic additive [13]. In addition, products derived from camelina are being used as ingredients in animal feed. Camelina cake, the by-product of the crushing process, is rich in protein (45% protein, 13% fiber, 10% oil residue, 5% vitamins and minerals), but if overused can have negative effects on livestock [14,15]. Additionally, due to high levels of linolenic acid and gamma-tocopherol, cold-pressed and unrefined camelina oil is well-suited for human nutrition. While generally low, the erucic acid content needs to be addressed by future breeding programs before camelina oil can be used as an edible oil in jurisdictions with more stringent food safety limits for this fatty acid [16].

Camelina has multiple industrial and feed applications and can offer a wide variety of benefits to producers and consumers. Considering that camelina breeding efforts only started at the beginning of the 21st century and in view of the large variability in terms of seed yield and yield-related traits, there is a tremendous potential for future improvement of camelina in present and future breeding programs [17–19].

The present study aimed to analyze the variability of seven yield-related traits of twenty cultivars, breeding lines and populations of spring camelina under different environmental conditions during a two-year field experiment. Herein, we report the categorization of the tested lines according to their yield stability and productive performance, and the most stable and productive lines will be identified. The effect of different environments (locations, seeding dates and weather conditions) on yield-related traits will be evaluated by measuring plant height (PH), height to the first branch (HFB), number of lateral branches (NLB), number of seed capsules per plant (NSCP), number of seeds per plant (NSP), mass of seeds per plant (MSP), and the total oil content of the seed (TPOS). The utility of the presented results for further improvement and selection of camelina genotypes more suitable for specific environmental conditions will be discussed.

## 2. Materials and Methods

### 2.1. Material

Twenty spring camelina accessions of different origin were used in the study (Table 1).

**Table 1.** Description of camelina accessions used in the study.

| Number | Accession | Origin | Description |
|--------|-----------|--------|-------------|
| 1 | NS Zlatka | IFVCNS collection (Serbia) | cv |
| 2 | NS Slatka | IFVCNS collection (Serbia) | cv |
| 3 | Maksimir | Croatia | cv |
| 4 | Maslomania | Ukraine | pop |
| 5 | CK2X-7 | BOKU (Austria) | bl |
| 6 | CK3X-7 | BOKU (Austria) | bl |
| 7 | CK2X-9 | BOKU (Austria) | bl |
| 8 | CK1X-25 | BOKU (Austria) | bl |
| 9 | CJ11X-43 | BOKU (Austria) | bl |
| 10 | CJ11X-79 | BOKU (Austria) | bl |
| 11 | Zavolzskij | BOKU (Austria) | cv |
| 12 | Omskij local | BOKU (Austria) | pop |
| 13 | Irkutskij local | BOKU (Austria) | pop |
| 14 | Pernice | BOKU (Austria) | cv |
| 15 | TYP Klagenfurt | BOKU (Austria) | pop |
| 16 | Leindotter Korneuburg | BOKU (Austria) | pop |
| 17 | Unkrautform Rollsdorf | BOKU (Austria) | pop |
| 18 | Gomholka | BOKU (Austria) | cv |
| 19 | Iwan | BOKU (Austria) | cv |
| 20 | Calena (fill) | BOKU (Austria) | cv |

cv—cultivar; bl—breeding line; pop—population.

### 2.2. Field Experiment and Design

A two-year study was carried out during the 2017–2019 growing seasons. Field trials were conducted at two locations: Rimski Šančevi (45°19′53″ N, 19°50′03″ E) and Bački Petrovac (45°20′04″ N, 19°40′16″ E), 12 km apart. The soil at both locations is chernozem with good humic horizon and a favorable air–water regime. Basic soil fertility parameters were measured before the first tillage in 2017 and are presented in Table 2.

**Table 2.** Soil fertility parameters for both locations at the beginning of the field trials.

| Location | pH | | CaCO$_3$ | Humus | Total N | AL-P$_2$O$_5$ | AL-K$_2$O |
|----------|--------|----------|----------|-------|---------|-----------|-----------|
| | in KCl | in H$_2$O | % | % | % | mg/100g | mg/100g |
| 1. Rimski Šančevi | 7.5 | 8.15 | 6.29 | 2.46 | 0.183 | 19.2 | 30 |
| 2. Bački Petrovac | 7 | 7.77 | 1.14 | 2.92 | 0.217 | 10.3 | 24.5 |

pH in KCl—active acidity–pH in H$_2$O; pH in H$_2$O—potential acidity–pH in 1M KCl; CaCO$_3$ content; humus content; total N—total content of nitrogen; AL-P$_2$O$_5$—available phosphorus; AL-K$_2$O—available potassium.

All soil analyses were performed in the Laboratory for Soil and Agroecology in the Institute of Field and Vegetable Crops, Novi Sad, Serbia. The active acidity–pH in H$_2$O was determined in a suspension (10 g:25 cm$^3$) of soil in water, and the determination of the potential acidity–pH in 1 M KCl was conducted in a suspension (10g:25 cm$^3$) of soil in potassium chloride, potentiometrically [20]. CaCO$_3$ content was determined with a calcimeter after Scheibler [20]. Humus content was determined by the Tjurin method [21]. Total content of nitrogen was determined by the total combustion method [22]. Available phosphorus was determined by the blue method in a spectrophotometer, and available potassium content was determined via flame photometer [23].

The same agronomic practices were applied at both locations. After the harvest of winter wheat (preceding crop in both years), main agronomic operations were carried out at the optimal time. The stubble was cultivated by disc harrowing, and the soil was prepared by shallow plowing (30 cm). After plowing, secondary tillage or pre-seeding soil preparation followed with a spike tooth harrow. Mineral fertilizer was applied at up to 300 kg ha$^{-1}$ NPK mineral fertilizer (15:15:15) at both sites. A fine-grained soil structure, needed for successful sowing of camelina, was achieved [17].

Herbicides and insecticides were applied under optimal weather conditions at rates recommended by the manufactures. Weed spraying occurred 3 weeks before seeding and in the 3–5 leaf stage of camelina. Pesticide applications were performed in winter and twice in spring for the control of *Phyllotreta* spp, (flea beetle), *Psylliodes chrysocephala* (cabbage stem flea beetle), *Brassicogethes aeneus* (pollen beetle), *Ceutorhynchus napi*, and *C. pallidactylus* (cabbage stem weevils).

The field trials were set up in a randomized complete block design (RCBD) with three replicates. Plots consisted of 10 rows with a length of 1.5 m, and the sowing was performed at a depth of 1–2 cm. Row spacing was 12.5 cm; plant spacing within rows was not controlled because sowing was done manually. The genotype NS-Zlatka was sown around the trials and served as border to reduce any external environmental influence on the experiment.

Genotype by environment interaction was evaluated using a combination of year, location, and seeding date (autumn and spring), resulting in eight combinations (hereinafter called "environments"): 2 years × 2 locations × 2 seeding dates (Table 3).

**Table 3.** Environments evaluated in the field trials.

|  | Location | Sowing Date | Harvest Date | Growing Season |
|---|---|---|---|---|
| E1 | Rimski Šančevi (site 1) | 2nd October | 23rd May | 2017–2018 (Y1A) |
| E2 | Bački Petrovac (site 2) | 3rd October | 28th May | |
| E3 | Rimski Šančevi | 3rd April | 4th July | 2017–2018 (Y1S) |
| E4 | Bački Petrovac | 4th April | 5th July | |
| E5 | Rimski Šančevi | 3rd October | 26th June | 2018–2019 (Y2A) |
| E6 | Bački Petrovac | 23rd October | 27th June | |
| E7 | Rimski Šančevi | 6th March | 2nd July | 2018–2019 (Y2S) |
| E8 | Bački Petrovac | 7th March | 3rd July | |

*2.3. Yield Trait Determination and Analyses*

In the study, seven yield-related traits were analyzed: plant height (PH), height to the first branch (HFB), number of lateral branches (NLB), number of seed capsules per plant (NSCP), number of seeds per plant (NSP), mass of seeds per plant (MSP), and the total percentage of oil in the seed (TPOS). Thirty plants (10 plants per replicate) were manually harvested at random from the middle rows (3rd–7th row) of the plots to avoid edge effects. Measurements were conducted on individual plants at the Department of Field and Vegetable Crops laboratory, Faculty of Agriculture, University of Novi Sad.

Plant height (PH) and height to the first branch (HFB) were measured as the distance between the stem base at ground level to the tip of the main raceme and the first lateral branch, respectively. NLB represents the number of primary lateral branches that originate from the main stem, or the total number of branches if no main stem could be identified. Secondary and tertiary branching was not considered. The number of seed capsules per plant (NSCP) was determined by counting. Seeds were cleaned from dust, seed pods, and other plant residues using sieves and weighed to determine mass of seeds per plant (MSP). The number of seeds per plant (NSP) was obtained by using an automatic seed counter (Contador, LCGC Bio Analytic Solutions LLP, Telangana, India). The total percentage of oil in the seed (TPOS) was determined by Maran Ultra Resonance NMR (Resonance

Instruments Ltd., Witney, UK) following the manufacturer's instructions and according to the ISO 10565 (1998) standard [24]. Seeds were pre-dried at 50 °C for 24 h to achieve a moisture content below 8%. The NMR reference standard was selected from a sample whose oil content had been determined by Soxhlet extraction. The analyses were carried out in the chemical laboratory of the Institute of Field and Vegetable Crops, Novi Sad, Serbia.

*2.4. Statistical Analysis*

The response of quantitative yield components to varied environmental conditions is determined by the additive main effects of genotype (G) and environmental conditions (E), as well as by the non-additive effects of genotype by environment (G × E) interactions [25]. The non-additive effect G × E interactions are well described by the additive main effects and multiplicative interaction model (AMMI), which is widely used in plant breeding. The AMMI model uses a combination of analysis of variance (ANOVA) to describe the additive effect of G and E and principal component analysis (PCA) to explain the multiplicative part of the G × E interactions [26]. The AMMI model (Equation (1)):

$$Y_{ge} = u + \alpha_g + \beta_e + \sum_{n=1}^{N} \lambda_n \xi_{gn} \eta_{en} + Qge \tag{1}$$

where $Y_{ge}$ is trait mean of genotype $g$ in environment $e$, $\mu$ is the grand mean, $\alpha_g$ is the genotypic mean deviation, $\beta_e$ is the environmental mean deviation, $N$ is the number of PCA axis retained in the adjusted model, $\lambda_n$ is the eigenvalue of the PCA axis $n$, $\xi_{gn}$ is the genotype score for PCA axis $n$, $\eta_{en}$ is the score eigenvector for PCA axis $n$, $Q_{ge}$ is the residual, including AMMI noise and pooled experimental error.

Because all of the monitored traits showed that their primary component IPCA1 was highly statistically significant for the effect of the G × E interactions, the results were presented using AMMI1 biplots. In these biplots, the additive main effects of G × E interactions were plotted on the *x*-axis in trait-specific units, and IPCA1 scores were plotted on the *y*-axis as the square root of the trait scores. All accessions to the right of the vertical line (grand mean of the trait scores) are accessions that have a higher score for the respective trait than the grand mean. All accessions on the left side of the vertical line are accessions that underperformed in the particular trait score. The position of the accessions relative to the horizontal IPCA1 axis indicates their contribution to the additive main effect, which means that accessions closer to the horizontal line are more stable for a particular trait, regardless of environments. In other words, genotypes and environments that are positioned around the *x*-axis show a low effect level of interaction and a wide adaptability to all environments. Environments and genotypes who possess IPCA1 scores with the same sign (+ or −) are bound by positive interactions, and scores with opposite signs by negative interactions [27–30].

The AMMI analyses were carried out with the software Genstat ver. 12 (VSNi, Hemel Hempstead, UK), and the F-test was used to determine the statistical significance of sources of variation.

The same software was used to calculate the correlations between all seven selected traits. In quantitative genetics, correlations between traits can explain the proportion of variance that traits share due to genetic and environmental effects on these traits. Trait interdependence and the correlation between traits were determined by computing the Pearson's correlation coefficient.

## 3. Results and Discussion

*3.1. Weather Conditions*

The trials were conducted during two growing seasons: 2017/2018 and 2018/2019. Both seasons had weather conditions typical for the semi-continental and semi-arid climate of the trial sites [31] (Figure 1). Monthly mean values for air temperature were similar at both locations and did not deviate significantly from the long-term averages. Contrast-

ingly, in certain months, mean precipitation values at the tested locations were drastically different from both long-term averages and each other. Two critical periods were identified: (i) extreme rainy and stormy weather in May and June 2018, which resulted in a delayed harvest of autumn-seeded camelina (Y1A) and caused substantial damage to both autumn- and spring-seeded plants, and (ii) extremely dry autumn and winter conditions in 2018/2019, which postponed germination in trials Y2A and Y2S.

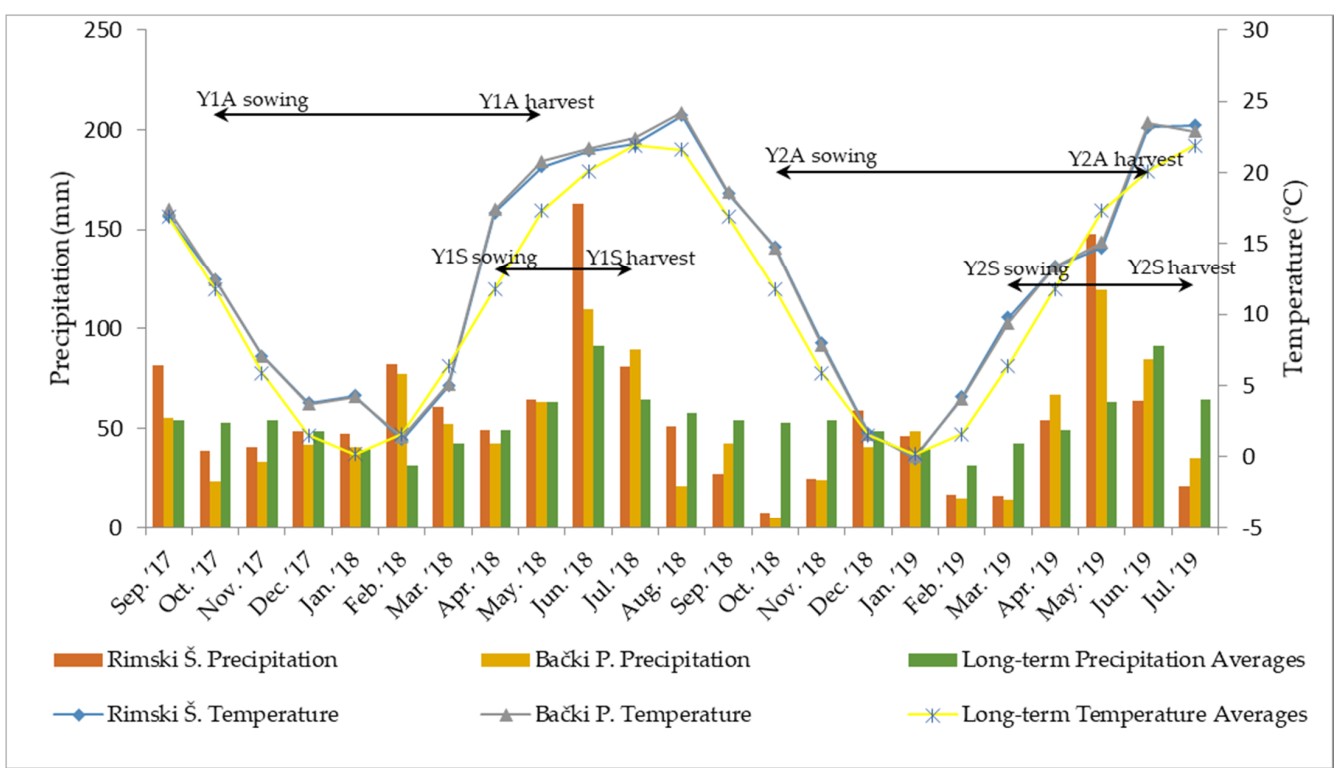

**Figure 1.** Weather conditions during the field trials, from September 2017 to July 2019. Columns represent monthly and long-term precipitation means, and lines represent monthly and long-term air temperature means. Y1A—first growing season, autumn-seeded camelina; Y1S—first growing season, spring-seeded camelina; Y2A—second growing season, autumn-seeded camelina; Y2S—second growing season, spring-seeded camelina.

### 3.2. Genotypes and Traits

The investigation and future development of promising genotypes, tolerant to diverse environmental conditions and, thus, more sustainable, are some of the most important and challenging tasks for plant breeders [32]. Knowing the extent to which genotype, environment, and G × E interactions affect a particular plant trait is of great importance for informing the selection of the next generation of genotypes. In this context, the adoption of AMMI analysis provides valuable information and allows for the estimation of genotype performance for all selected traits (Table 4).

Overall, both genotype and environment effects proved to be significant for all traits. The influence of environmental factors contributed to the largest proportion of the total variance, explaining approximately 90% of the variance for all yield traits (except NLB with 70.96% and TPOS with 42.56%). Genotype explained 1.84–6.45% of the total variance (except TPOS with 35.51%). G × E interactions explained 3.6–6.35% of the variance for all selected traits (except NLB with 22.58% and TPOS with 21.92%). The first principal component, IPCA1, was significant for all tested traits, explaining from 28.04% (NSCP) to 56.64% (MSP) of variation. IPCA2 was not significant only for NSP and MSP, for all other traits it explained from 19.22% (NLB) to 30.01% (TPOS) of variation. The third principal

component was significant only for PH and HFB and explained from 11.26% (NLB) to 20.18% (NSP) of variation.

**Table 4.** Additive main effect and multiplicative interaction (AMMI) analysis of variance in traits of camelina genotypes across eight environments.

| Source of Variation | df | PH | | HFB | | NLB | | NSCP | | NSP | | MSP | | TPOS | |
|---|---|---|---|---|---|---|---|---|---|---|---|---|---|---|---|
| | | F | VE (%) | F | VE (%) | F | VE (%) | F | VE (%) | F | VE (%) | F | VE (%) | F | VE (%) |
| Treatments | 159 | 50.21 ** | 100 | 21.29 ** | 100 | 6.74 ** | 100 | 21.65 ** | 100 | 17.85 ** | 100 | 25.65 ** | 100 | 5.32 ** | 100 |
| Genotypes (G) | 19 | 11.21 ** | 2.67 | 3.85 ** | 2.16 | 3.64 ** | 6.45 | 4.88 ** | 2.69 | 4.32 ** | 2.89 | 3.96 ** | 1.84 | 15.8 ** | 35.51 |
| Environments (E) | 7 | 219.77 ** | 93.73 | 158.28 ** | 91.49 | 24.42 ** | 70.96 | 188.82 ** | 92.3 | 147.02 ** | 91.51 | 371.56 ** | 93.49 | 3.78 ** | 42.56 |
| G × E | 133 | 2.16 ** | 3.6 | 1.62 ** | 6.35 | 1.82 ** | 22.58 | 1.3 * | 5.0 | 1.19 | 5.59 | 1.43 * | 4.65 | 1.4 * | 21.92 |
| IPCA 1 | 25 | 4.52 ** | 39.3 [1] | 2.96 ** | 34.46 [1] | 4.38 ** | 45.25 [1] | 1.93 ** | 28.04 [1] | 3.43 ** | 53.99 [1] | 4.29 ** | 56.64 [1] | 2.68 ** | 36.23 [1] |
| IPCA 2 | 23 | 2.81 ** | 22.5 [1] | 2.14 ** | 22.85 [1] | 2.02 ** | 19.22 [1] | 1.65 * | 22.04 [1] | 1.42 | 20.51 [1] | 1.34 | 16.08 [1] | 2.42 ** | 30.01 [1] |
| IPCA 3 | 21 | 1.84 * | 13.45 [1] | 1.99 ** | 19.43 [1] | 1.3 | 11.26 [1] | 1.49 | 18.12 [1] | 0.82 | 20.18 [1] | 0.82 | 12.23 [1] | 1.29 | 14.56 [1] |
| Residuals | 64 | 1.11 | 24.74 | 0.78 | 23.27 | 0.92 | 24.27 | 0.86 | 31.79 | 0.36 | 27.03 | 0.45 | 15.03 | 0.56 | 19.18 |
| Error | 304 | / | / | / | / | / | / | / | / | / | / | / | / | / | / |

[1] percentage of variance in G × E interactions; PH—plant height; HFB—height to the first branch; NLB—number of lateral branches; NSCP—number of seed capsules per plant; NSP—number of seeds per plant; MSP—mass of seeds per plant; TPOS—total percentage of oil in the seed; df—degrees of freedom; F—F-values; VE—variation explained; IPCA1–3—principal components; * and **—statistical significance for $p \leq 0.05$ and $p \leq 0.01$.

All investigated traits were presented separately on AMMI-1 biplots in order to identify genotypes with the highest mean values for the selected trait and the highest stability under different environmental conditions. Correlations between the seven yield traits are presented in Table 5. Identifying correlations between yield traits can be very important for plant breeders, particularly when selecting traits on which future selections will be based.

**Table 5.** Pearson correlation coefficient ($r$) between yield-related traits for all trials conducted during 2017–2019.

| | **Pearson Correlation Coefficients ($r$)** | | | | | | |
|---|---|---|---|---|---|---|---|
| PH | / | | | | | | |
| HFB | 0.42 ** | / | | | | | |
| NLB | 0.54 ** | −0.08 ns | / | | | | |
| NSCP | 0.79 ** | 0.02 ns | 0.62 ** | / | | | |
| NSP | 0.82 ** | 0.11 * | 0.65 ** | 0.92 ** | / | | |
| MSP | 0.82 ** | 0.12 ** | 0.62 ** | 0.93 ** | 0.95 ** | / | |
| TPOS | −0.24 ** | −0.32 ** | −0.17 ** | −0.17 ** | −0.17 ** | −0.22 ** | / |
| | PH | HFB | NLB | NSCP | NSP | MSP | TPOS |

* and **—statistical significance for $p \leq 0.05$ and $p \leq 0.01$; ns = not significant.

### 3.2.1. Plant Height (PH)

Plant architecture is an important determinant of photosynthetic efficiency and, thus, overall crop performance. Optimization of plant architectural traits through selection can have great potential for increasing biomass and/or seed yield. One of the most crucial plant architecture traits is PH [33]. In the present study, PH showed a positive correlation with all other traits (Table 5). The strongest correlations were observed with NSCP, NSP, and MSP, which indicates that taller camelina plants achieved higher seed yields. These observations agree with those made in other studies [25,26,34].

The AMMI1 biplot for average PH (Figure 2A) shows that genotypes were on average taller during the second growing season (E5–E8). In particular, G1, G11, and G16 were the tallest genotypes with mean values of 112.47, 110.3, and 109.2 cm, respectively. The highest average PHs were identified at Rimski Šančevi in the second year (Table S1). Sowing time did not affect the variation as much as years and the agroecological conditions of the two locations did. Over the entire experiment, E5 showed significantly higher values for almost all surveyed traits, likely in response to irrigation that was applied in autumn, which

promoted germination and plant establishment during the prolonged drought period. The application of irrigation allowed for good stand establishment, which contributed to the highest PH values for that year, confirming how important the establishment phase is for camelina yield production, with its associated water requirements [35,36].

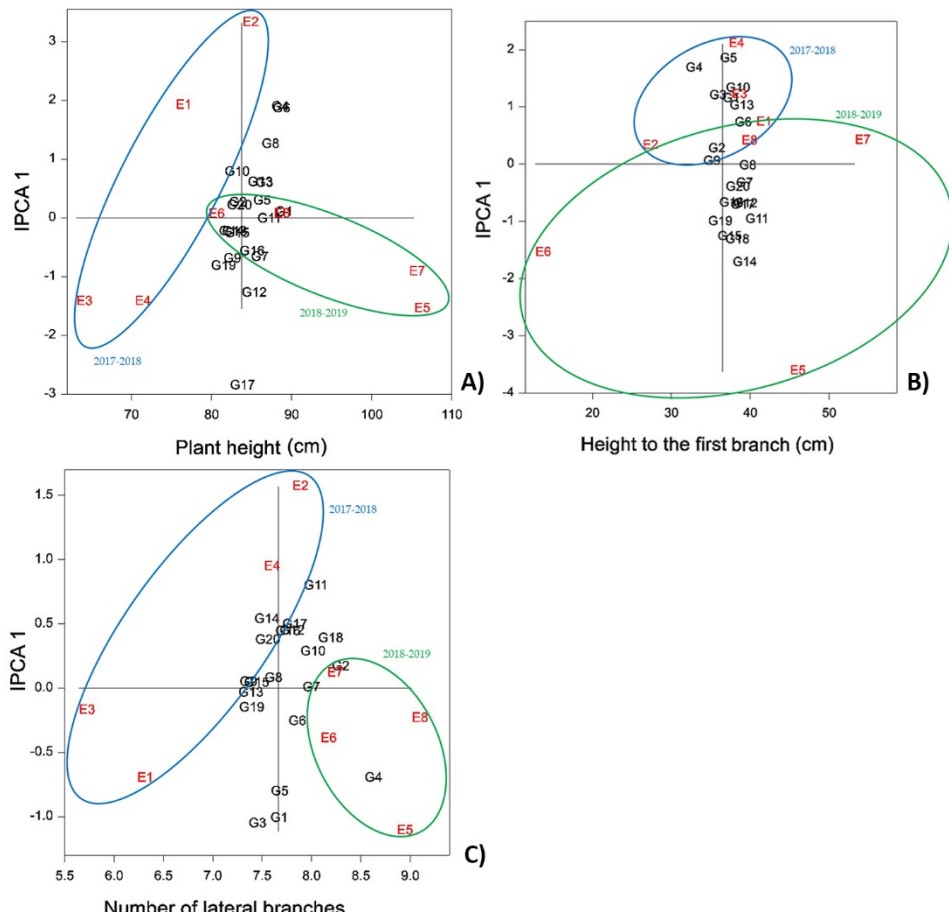

**Figure 2.** AMMI 1 biplots for (**A**) plant height (PH), (**B**) height to the first branch (HFB), and (**C**) number of lateral branches (NLB) of 20 camelina accessions across eight experimental environments. E = environment; G = genotype; blue frame—first growing year 2017–2018; green frame—second growing season 2018–2019.

In both growing seasons, camelina genotypes achieved higher average PHs when seeded in the autumn (Table S1). A longer vegetation phase allowed camelina plants to produce more biomass, resulting in higher yields (positive correlation between the two traits—Table 5). In addition, autumn establishment allowed camelina to be more competitive against weeds in spring [37]. Lines G1, G11, and G5 showed the greatest stability over all environmental factors and reached higher PHs than the grand mean, so they should be considered for future selections for this trait. Furthermore, lines G4, G6, and G8 showed very high overall PH scores, but the large degree of score variability indicates limited adaptability to different environmental conditions. Future selections of camelina that might be done by PH should be done very carefully, as taller plants are more prone to lodging, especially under high N regimes of production. Therefore, breeders should find the right balance between the plant height and the available N in the soil.

3.2.2. Height to the First Branch (HFB)

HFB can to some extent be used to determine which branching pattern a camelina genotype possesses. In the study [25], four branching patterns were identified (W, X, Y, and Z-type). Top heavy branching (W-type) with secondary branching was the most common

for camelina plants in this study. The X-type was the second most common, with lateral branching throughout the whole length of the main stem and little secondary branching. Y-type plants were characterized by lateral branches that originate from the stem base and little secondary branching, and the least common were Z-type plants, with lateral branches emerging everywhere, seemingly without apical dominance.

HFB may be a very important trait, as it is one of the main factors (next to PH) that determines the thickness of the seed canopy and, thus, yield. It is also a trait that determines the distance of the seed canopy from the ground level, i.e., clearance. This is an important trait for combinability of the crop, as seed pods that are too close to the ground cannot be combined. Even if the seed pod shattering is still the main factor for losses caused by mechanized harvesting [38,39], future breeders should look for the best combination of PH, thickness of seed canopy, and clearance as a possibility to reduce these losses.

During the present study, genotype and environment explained 91.49% of the total variance for HFB. This trait is the only one not significantly correlated with other yield components, except for a weak negative correlation with TPOS (−0.32, Table 5). AMMI 1 analysis for HFB showed that overall, higher mean values were observed during the second growing season, but unlike PH, this trait was much more responsive to the differences in agroecological conditions that occurred at the two locations and for the different sowing times (Figure 2B).

Considering data on the average HFB (Table S2) and the positioning of the environments (Figure 2B), it is possible to draw the conclusions that spring sowing and location 1 (Rimski Šančevi) resulted in branching occurring much higher on the main stem. The highest values for HFB were reported during the second year (spring sowing) at location 1 for G15, G8, and G1, reporting values of 57.13, 56.33, and 55.97 cm, respectively. These genotypes most likely belong to the W-type. Otherwise, the genotypes G2, G4, and G9 showed the lowest values of 9.2, 5.9, and 9.57 cm, respectively, in the second year at location 2 (Bački Petrovac) with autumn sowing. Thus, these genotypes most likely belong to the Y- or Z-type. For accurate determination of the genotype branching pattern, plants should be grown under controlled conditions where they can freely express their genetic potential without any interference from the environment [25].

In terms of stability for HFB, all genotypes showed relatively stable values except G4, G5, G14, and G18. This indicates that the branching pattern of the latter is susceptible to different environmental conditions and plant densities.

### 3.2.3. Number of Lateral Branches (NLB)

NLB is an important trait for breeding purposes, as it directly influences total seed yield in camelina [40]. This statement is supported by the results of the present study, as NLB showed strong positive correlations with NSCP, NSP, and MSP (Table 5). Regarding the additive effect, G and E accounted together for 77.41% (E = 70.96%, G = 6.45%), while G × E interactions accounted for 22.58% of the phenotypic variation. The first principal component, IPCA1, explained 45.25% of the total variability in interactions for NLB (Table 4).

The positioning of environments and genotypes in the AMMI 1 biplot for NLB (Figure 2C) indicates that the average scores for NLB were higher during the second year. This means that the lines generally achieved greater biomass during the second year, with direct positive effects on NSCP, NSP, and MSP. The different agroecological conditions at the two sites significantly affected NLB as well, with greater branching at the second location. Over both growing seasons, plants seeded in autumn produced more branches than plants seeded in spring. The same results were achieved during the [41] study.

Lines G2, G4, G7, and G18 in E8 (second trial year, location 2, spring-seeded) had the highest number of lateral branches with 11.07, 10.93, 10.37, and 9.7, respectively (Table S3). During our study, plant density was another factor that influenced NLB. Because of the heavy drought during autumn and winter of 2018/19, the germination was uneven, and as a consequence, plants had more space to develop, producing more biomass and increasing

their NLB. Thus, in addition to water regime and sowing date, plant density played a role in promoting higher average scores for NLB, NSCP, NSP, and MSP during the second growing year. Similar conclusions were achieved during [42]'s study.

The camelina genotypes G2 and G7 showed the highest stability for NLB over all environments, and their scores were higher than the grand mean. Thus, genotypes G2, G4, G7, and G18 could be interesting for future breeding selections based on NLB.

### 3.2.4. Number of Seed Capsules per Plant (NSCP), Number of Seeds per Plant (NSP), Mass of Seeds per Plant (MSP)

NSCP, NSP, and MSP are traits that strongly influence the overall seed yield of camelina. All three are highly correlated (Table 5), as observed by [43]. Environment explained the largest proportion of the variance for all three traits at 91.51 to 93.49%. IPCA1 for NSP and MSP explained 53.99–56.64% of the G × E interactions, and NSCP had two relevant principal components (Table 2). As reported earlier, the environmental conditions of the second year (2018–2019) resulted in greater results for all three traits. Average values surveyed for the 20 accessions during the first year were NSCP = 117.56, NSP = 884.67, and MSP = 0.79 g, and during the second year, they were NSCP = 236.7, NSP = 2635.5, and MSP = 2.46 g (Tables S4–S6), indicating that the number of seeds per seed capsule increased on average by almost 50% (7.5 vs. 11) in the second year. The main effects and their interactions are presented in Figure 3.

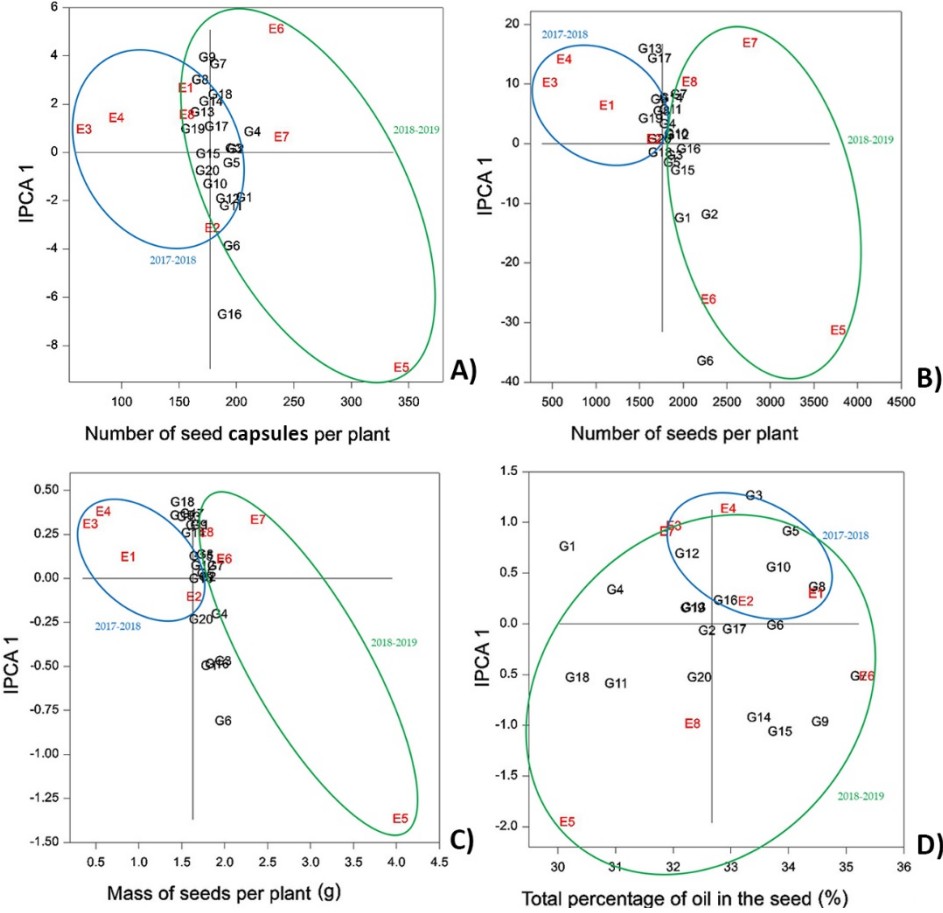

**Figure 3.** AMMI 1 biplots for (**A**) number of seed capsules per plant (NSCP); (**B**) number of seeds per plant (NSP); (**C**) mass of seeds per plant (MSP); (**D**) total percentage of oil in the seed (TPOS) of 20 camelina accessions across eight experimental environments. E = environment; G = genotype; blue frame—first growing year 2017–2018; green frame—second growing season 2018–2019.

Weather conditions had the greatest impact on NSCP, NSP, and MSP, while the micro-climate at the two locations did not have a significant effect: site 2 (Bački Petrovac) showed overall better results in the first year, and site 1 (Rimski Šančevi) showed better results during the second growing season. For all three traits, the results for E5 showed both a large interaction effect and high average performance. This can be explained by the irrigation that was applied in autumn of the second year [44]. Autumn seeding generally provided better results for NSCP, NSP, and MSP in comparison to spring sowing, as has been observed in other studies [5,34,35,45].

For NSCP, lines G16, G6, and G1 reported the highest values in both seasons with averages of 392.63, 391.13, and 382.47, respectively. At the same time, G1 and G4 had the highest stability over all environments (Figure 3A).

Concerning NSP, again lines G6, G16, and G1 were the best performing, while G16, G12, and G10 were most stable for this trait (Figure 3B).

MSP showed similar genotype distribution on the AMMI 1 biplot as NSCP and NSP. Again, G6, G16, and G1 reported the highest values, along with G3, corresponding to 5.29, 4.80, 4.70, and 4.94 g, respectively. The most stable genotypes were G3, G4, and G7 (Figure 3C).

In summary, lines G1, G6, and G16 appear as the most promising genetic material for use in breeding programs to improve NSCP, NSP, and MSP.

### 3.2.5. Total Percentage of Oil in the Seed (TPOS)

Camelina is a rediscovered oilseed crop; therefore, total seed oil content is one of the most important breeding objectives. It is well known that quality and quantity of the seed oil are affected by environmental factors [45]. Of the characterized traits, TPOS was the only trait that was strongly determined by genotype. Genotype explained 35.51% of the variance, while environment and their interaction explained 42.56 and 21.92%, respectively (Table 2).

The average TPOS in this study was 32.67%. Most genotypes and environments showed high interaction effects, indicating low stability and adaptability (Figure 3D). This is to be expected for traits that are highly influenced by genetic differences. Sum and distribution of precipitation between locations showed little effect on TPOS, as in the first-year, site 1 showed better results, and site 2 showed better results in the second year. Contrastingly, seeding date had a significant effect on TPOS with higher seed oil contents in autumn-seeded camelina in both years (31.66 vs. 34.05%). The same results were achieved by numerous other studies in camelina [4,5,45,46].

The highest stability for TPSO over all environments was observed for G6 and G17 (Table S7). Accessions G7, G9, G8, G5, and G15 possessed the highest average seed oil contents, making them good candidates for use in future breeding programs; however, it needs to be taken into account that G9, G5, and G15 were less stable for this trait.

### 3.3. Effect of the Environments

By means of the AMMI statistical model, it was possible to obtain reliable information about the adaptability and stability of each line and analyze how different environmental conditions affected yield-related traits. Averaged over all seven traits, variation was explained mostly by environmental conditions (82.26%), followed by genotype × environment interactions (9.95%) and genotypes (7.74%).

Weather conditions were the main factor influencing all of the traits. TPOS was the only trait that was equally affected by the genetic differences between the accessions and the weather conditions. Similar observations have been made by [5,26]. While temperature and precipitation means were similar to the long-term averages, precipitation distribution appeared to be the main meteorological factor with that influencing the expression of the studied traits. Extreme rainy and stormy weather, during the harvest of camelina in the first growing season 2017–2018, was one of the main factors that made a big difference in trait values between the two seasons. Strong rainfall and harsh winds made substantial damage

to the ripe plants making the seed pots shatter and the whole plants log down. Somewhere between 30–40% of yield was lost. Contrary to that dry autumn and winter conditions are also a major problem in our climate conditions. The drought conditions postponed camelina germination in the second year. If irrigation was not applied during the autumn on site 1 (Rimski Šančevi), there is a big possibility that the crop would be completely lost. One additional factor that should be carefully considered is the presence of the Turnip Yellow Virus (TuYV). The presence of TuYV was visually noted and later confirmed by molecular analysis of sampled plant material, in both years (Figure S1). The wet and warm weather conditions of the first growing season were ideal for the virus to spread and cause significant damage to the plants, as has been reported in [47]. TuYV infection caused approximately 20.62% of yield damage during the first growing season, making the plants stunted and pale and reducing their branch number, seed pod count, and seed count in the pods. According to [25], dryer years seem to be more favorable for camelina yield production, while humid conditions promote diseases and make problematic conditions during harvest more likely.

The second environmental factor considered in the present study was sowing time. During both years, autumn seeding positively affected the highly correlated traits PH, NLB, NSCP, NSP, and MSP (Table 5). This confirmed that spring camelina genotypes can easily withstand winter temperatures (only seldom below 0 °C) and drought conditions, which occur frequently in the study region. It also shows that the longer vegetation time, in good years, leads to greater plant biomass, which is generally correlated with increased seed yield [4,5,40,46].

As a third factor, two locations were compared. Even though the sites had similar soil characteristics and fertility parameters and are only 12 km apart, it was possible to notice that their agroecological conditions differed enough to significantly influence camelina growth and yield. As to be expected, the largest differences were observed during the second growing season with the application of irrigation at site 1 (Rimski Šančevi). This agronomic practice made E5 the most suitable environment for growing camelina. Nevertheless, apart from the irrigation, the main differences between the two locations were related to the differences in weather conditions (sum and distribution of precipitation).

*3.4. Genotype Ranking*

The goal of this study was to characterize 20 spring camelina accessions in relation to yield traits as well as stability and adaptability to different environmental conditions. Based on the average means for the seven traits (Tables S1–S7) and the effects of different environmental conditions (Figures 2 and 3), all accessions were ranked for each trait (Table 6). More details can be found in Figure S2. The ranking (rank I being the best) presented here includes the four best accessions, which might be considered for future breeding.

**Table 6.** Ranking of the 20 spring camelina accessions by their average trait means and their stability in various environments.

| Yield Trait | Ranking of Accessions | | | |
|:---:|:---:|:---:|:---:|:---:|
| | I | II | III | IV |
| PH | G1 | G6 | G4 | G8 |
| HFB | G11 | G14 | G6 | G8 |
| NLB | G4 | G2 | G18 | G7 |
| NSCP | G4 | G1 | G2 | G6 |
| NSP | G6 | G2 | G16 | G1 |
| MSP | G6 | G3 | G16 | G4 |
| TPOS | G7 | G9 | G8 | G5 |

PH—plant height; HFB—height to the first lateral branch; NLB—number of lateral branches; NSCP— number of seed capsules per plant; NSP—number of seeds per plant; MSP—mass of seeds per plant; TPOS—total percentage of oil in the seed.

The phenotypic evaluation and the statistical analysis allowed the identification of accessions with promising genetic potential for different yield-related traits. G6 (CK3X-7) and G1 (NS Zlatka) proved to be one of the most suitable lines for the test conditions, performing well for all selected traits. The breeding line G6 (CK3X-7) showed the highest seed yield potential, as demonstrated by a high average number of seeds per plant of up to 5056 and high average mass of seeds per plant (maximum MSP = 5.29 g). G3 (Maksimir) showed second best seed yield potential with 4.94 g MSP. These two genotypes may be considered as good starting material for future breeding programs aiming at developing camelina cultivars with high seed yield and good yield stability. In addition, G7 (CK2X–9) and G9 (CJ11X–43) may be used in future breeding efforts for the improvement of total seed oil content.

As the environments contributed the largest proportion of the total variance, the rankings of all 20 spring camelina accessions for every trait depending on the different sowing dates are presented (Table S8). These rankings may prove to be of great utility to breeders, as sowing date was the only agronomic factor that had a major influence on the study results.

## 4. Conclusions

Climate change is one of the main threats to future agricultural production. The task of identifying crops and genotypes within crops that are more resilient to climate change represents a real challenge for agriculture in general and plant breeders in particular. Because camelina is a crop that possesses broad environmental adaptability, requires few inputs, and is resistant to many diseases and pests, this ancient and almost forgotten oilseed crop may have an important place in the future portfolio of oilseed crops. The presented study confirmed that the expression of yield-related traits in camelina is mainly driven by environmental conditions, even if some of the tested genotypes showed higher performance and lower variability than others. This allowed identifying some camelina lines, such as G3 (Maksimir) and G6 (CK3X-7), which could be considered potential good candidates for improving seed yield, but also others, such as G7 (CK2X–9) and G9 (CJ11X–43), are highly suitable for improving seed oil content. The results of this study may assist future breeding programs in the development of higher-yielding cultivars with increased seed oil content and greater adaptability to various environmental conditions.

**Supplementary Materials:** The following are available online at https://www.mdpi.com/article/10.3390/agronomy11050858/s1. Table S1. Average PH of camelina accessions for two growing seasons (2017/18 and 2018/19); Table S2. Average HFB of camelina accessions for two growing seasons (2017/18 and 2018/19); Table S3. Average NLB of camelina accessions for two growing seasons (2017/18 and 2018/19); Table S4. Average NSCP of camelina accessions for two growing seasons (2017/18 and 2018/19); Table S5. Average NSP of camelina accessions for two growing seasons (2017/18 and 2018/19); Table S6. Average MSP of camelina accessions for two growing seasons (2017/18 and 2018/19); Table S7. Average TPOS of camelina accessions for two growing seasons (2017/18 and 2018/19); Table S8. Ranking of the 20 spring camelina accessions by their average trait means at two different sowings dates (autumn and spring); Figure S1: Presence of Turnip yellow virus (TuYV) in different vegetation periods: (A) Location 1 (Rimski Šančevi) winter sowing of first growing season (2017–2018); (B) Location 2 (Bački Petrovac) spring sowing of second growing season (2018–2019); Figure S2. GGE ranking biplots for all camelina accessions for each trait for two growing seasons (2017/18 and 2018/19); (a) GGE ranking biplot for PH; (b) GGE ranking biplot for HFB; (c) GGE ranking biplot for NLB; (d) GGE ranking biplot for NSCP; (e) GGE ranking biplot for NSP; (f) GGE ranking biplot for MSP; (g) GGE ranking biplot for TPOS.

**Author Contributions:** Conceptualization, B.K., A.M.J. and V.M.; methodology, A.M.J., S.P. and N.N.; software, B.K. and V.M.; validation B.K., A.M.J. and N.N.; formal analysis, B.K. and N.G.; investigation, B.K., A.M.J. and V.M.; resources J.V., A.M.J., S.P. and V.M.; data curation, B.K. and A.M.J.; writing—original draft preparation, B.K.; writing—review and editing, B.K., A.M.J., V.M., N.N., F.Z., C.E., J.V. and S.P.; visualization, B.K., S.P., V.M. and A.M.J.; supervision, S.P., A.M.J. and

F.Z.; project administration, B.K., A.M.J. and V.M.; funding acquisition, B.K. All authors have read and agreed to the published version of the manuscript.

**Funding:** This research received no external funding.

**Data Availability Statement:** Data sharing not applicable.

**Acknowledgments:** Authors want to thank The Institute of Field and Vegetable Crops for providing all necessary resources for conducting this research; Faculty of Agriculture, The University of Novi Sad for providing laboratory support, and Johann Vollmann (Institute of Plant Breeding, University of Natural Resources and Life Sciences—BOKU, Vienna, Austria) for providing the spring camelina genotype seeds.

**Conflicts of Interest:** The author Boris Kuzmanovic is an employee of MDPI; however, he does not work for the journal Agronomy at the time of submission and publication.

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
