# Peer review of "Yield-Related Traits of 20 Spring Camelina Genotypes Grown in a Multi-Environment Study in Serbia"

_agronomy, doi:10.3390/agronomy11050858_

Round 1

Reviewer 1 Report

The article concerns the analysis of plant traits important from the point of view of plant breeding. This topic is rarely discussed in today's publications.

The inclusion of various accessions, including breeding lines of camelina in the experiment, makes the publication up-to-date.

The weak point of the work is the short period of the experiment (2 years) and the location of two experimental points (with similar conditions) next to each other. The data presented in the study show, that they differed practically only in the sum of precipitation.

These issues make it difficult to draw conclusions or make recommendations based on experience. It is especially important when they are associated with additional difficulties, such as weather anomalies (rainstorms, droughts). It is good that the authors recognize these problems and take into account some of them in the discussion.

Most of the plant characteristics analyzed in this study are closely related to the plant density per unit area, which shapes the plant habit. I propose to include a table with this feature and discuss it briefly. This would support the discussion in section 3.2.3, lines 336-342.

In chapter 3.2.2 the Authors refer to 4 camelina branching patterns. Is it reasonable to infer the genotype type from just one feature (height to the first branch) - lines 311 – 314?

In the manuscript, I encountered some errors or inaccuracies that I refer directly to in the text.

Author Response

Dear Reviewer,

Thank you very much for your consideration and invaluable comments to our recent submission. Your suggestions and feedback on the content were helpful in our revision. We appreciate your effort and time. We carefully went through all of your comments and revised the manuscript accordingly. We have revised, taking all your comments into account, added content and re-organized the writing.

General comments: The article concerns the analysis of plant traits important from the point of view of plant breeding. This topic is rarely discussed in today's publications.

The inclusion of various accessions, including breeding lines of camelina in the experiment, makes the publication up-to-date.

The weak point of the work is the short period of the experiment (2 years) and the location of two experimental points (with similar conditions) next to each other. The data presented in the study show, that they differed practically only in the sum of precipitation.

These issues make it difficult to draw conclusions or make recommendations based on experience. It is especially important when they are associated with additional difficulties, such as weather anomalies (rainstorms, droughts). It is good that the authors recognize these problems and take into account some of them in the discussion.

[Response]: We do agree that the similarity between the two locations was high, but in analyzing the results of the study, we can conclude that the differing weather conditions at the two locations were an important factor for camelina trait expression. Even if the distance was short (12km), the difference between the two microclimates permitted us to do a complete characterization of the genetic material.

Comment 1: Most of the plant characteristics analyzed in this study are closely related to the plant density per unit area, which shapes the plant habit. I propose to include a table with this feature and discuss it briefly. This would support the discussion in section 3.2.3, lines 336-342.

[Response] The sowing of the plots was carried out manually and the exact plant density was not measured. Therefore, we are not able to provide the exact plant density for each plot of the experiment. We agree that the plant density has an influence on the selected traits, and this is why we harvested plants from the 3rd–7th row, as explained in lines 151–153 in order to avoid any edge effects. When comparing the rankings of the accessions between the two years, we can see that they are not aligned and that the joint effect of all factors had an impact on obtaining different rankings between the two years. As the plant density was not measured, we cannot estimate the importance and the exact influence of plant density on all traits. This is one of the limiting factors of this study. However, despite this, we were able to identify genotypes (G2 and G7) with high stability for NLB over all environments and scores that were higher than the grand mean. These lines will be extremely useful in future breeding efforts. In fact, one of these lines (G2) is an established commercial cultivar with high seed yield potential, which supports the favorable results obtained for NLB for this line in the present study.  

Comment 2: In chapter 3.2.2 the Authors refer to 4 camelina branching patterns. Is it reasonable to infer the genotype type from just one feature (height to the first branch) - lines 311 – 314?

[Response] We agree with your statement, and that it is not correct to presume the genotype branching type with only one feature like the height to the first branch. This is why we stated that it is most likely that the genotypes that have a high or a low HFB belong to one or two specific plant types. A more accurate explanation is given in lines 314–316.

Comment 3: In the manuscript, I encountered some errors or inaccuracies that I refer directly to in the text.

[Response] Thank you for pointing out these errors, we have revised all of the suggestions accordingly.

Reviewer 2 Report

The manuscript included evaluation adaptation of Camelina genotype to different whether, soil and seeding time. For this purpose, the classical AMMI method was used to assess G xE interactions. However, it seems not very correct to use a combination of three factors (location x years x sowing date) as the environments. This will certainly make difficult in correct interpretation of the results and it is uncertain whether the assumptions AMMI and ANOVA approaches. It is commonly found in the literature that he treats the environment as a combination of max two factors - locations x cultivation seasons (years). If we are dealing with a three-factor combination, then such inference is most often carried out separately for the cultivation factor. I recommend taking a look at the numerous literature dealing with the three-factor classification:

Laidig, F., H.P. Piepho, T. Drobek, and U. Meyer. 2014. Genetic and non-genetic long-term trends of 12 different crops in German official variety performance trials and on-farm yield trends. Theor. Appl. Genet. 127:2599–2617. doi:10.1007/s00122-014-2402-z

Studnicki, M., Paderewski, J., Piepho, H.P. and Wójcik‐Gront, E. (2017), Prediction Accuracy and Consistency in Cultivar Ranking for Factor‐Analytic Linear Mixed Models for Winter Wheat Multienvironmental Trials. Crop Science, 57: 2506-2516. https://doi.org/10.2135/cropsci2017.01.0004

Laidig F, Piepho HP, Rentel D, Drobek T, Meyer U, Huesken A. Breeding progress, variation, and correlation of grain and quality traits in winter rye hybrid and population varieties and national on-farm progress in Germany over 26 years. Theor Appl Genet. 2017 May;130(5):981-998. doi: 10.1007/s00122-017-2865-9. Epub 2017 Mar 13. PMID: 28289803; PMCID: PMC5395587.

Buntaran, H., Piepho, H.‐P., Hagman, J. and Forkman, J. (2019), A Cross‐Validation of Statistical Models for Zoned‐Based Prediction in Cultivar Testing. Crop Science, 59: 1544-1553. https://doi.org/10.2135/cropsci2018.10.0642

Here are some minor remarks:

Line 30-32 - the conclusion is true on the basis of the AMMI analysis performed

Line 204-207 - the pattern is so obvious, that it should be omitted

Table 4 - it would be more interesting to determine the significance of the sowing time effect and not such a combined environmental effect (I suspect that a large impact of the trait variability is caused by nesting sowing time in environmental effect).

Line 447 - I am curious if there is a compatibility

in the ranking of genotype between the study traits, as well as between the two sowing dates for individual trait. 

Author Response

Reviewer 2.

Dear Reviewer,

Thank you very much for your consideration and invaluable comments to our recent submission. Your suggestions and feedback on the content were helpful in our revision. We appreciate your effort and time. We carefully went through all of your comments and revised the manuscript accordingly. We have revised, taking all your comments into account, added content and re-organized the writing.

Comment 1: The manuscript included evaluation adaptation of Camelina genotype to different whether, soil and seeding time. For this purpose, the classical AMMI method was used to assess G xE interactions. However, it seems not very correct to use a combination of three factors (location x years x sowing date) as the environments. This will certainly make difficult in correct interpretation of the results and it is uncertain whether the assumptions AMMI and ANOVA approaches. It is commonly found in the literature that he treats the environment as a combination of max two factors - locations x cultivation seasons (years). If we are dealing with a three-factor combination, then such inference is most often carried out separately for the cultivation factor. I recommend taking a look at the numerous literature dealing with the three-factor classification:

Laidig, F., H.P. Piepho, T. Drobek, and U. Meyer. 2014. Genetic and non-genetic long-term trends of 12 different crops in German official variety performance trials and on-farm yield trends. Theor. Appl. Genet. 127:2599–2617. doi:10.1007/s00122-014-2402-z

Studnicki, M., Paderewski, J., Piepho, H.P. and Wójcik‐Gront, E. (2017), Prediction Accuracy and Consistency in Cultivar Ranking for Factor‐Analytic Linear Mixed Models for Winter Wheat Multienvironmental Trials. Crop Science, 57: 2506-2516. https://doi.org/10.2135/cropsci2017.01.0004

Laidig F, Piepho HP, Rentel D, Drobek T, Meyer U, Huesken A. Breeding progress, variation, and correlation of grain and quality traits in winter rye hybrid and population varieties and national on-farm progress in Germany over 26 years. Theor Appl Genet. 2017 May;130(5):981-998. doi: 10.1007/s00122-017-2865-9. Epub 2017 Mar 13. PMID: 28289803; PMCID: PMC5395587.

Buntaran, H., Piepho, H.‐P., Hagman, J. and Forkman, J. (2019), A Cross‐Validation of Statistical Models for Zoned‐Based Prediction in Cultivar Testing. Crop Science, 59: 1544-1553. https://doi.org/10.2135/cropsci2018.10.0642

[Response] Our study assessed G x E interactions using a 3-way AMMI method, the use of which has been published previously (see below). We do agree with you that the classical 2-way AMMI would not be appropriate for the interpretation of three factors (location x years x sowing date) as environments. This is why we used a 3-way AMMI, which has proven to provide useful information beyond what would have been available to the breeder through a 2-way AMMI. 3-way interaction effects cannot be detected by the 2-way AMMI analysis and some 2-way interactions are spurious because loss of information occurs when condensing modes. Therefore, results of the analysis of the 2-way interactions obtained by combining levels of factors do not seem to be appropriate; the analysis of the 3-mode data provides a complete and useful overview of the different interactions occurring in the trial.

Literature:

Varela, M., Crossa, J., Rane, J., Joshi, A.K., Trethowan, R. Analysis of a three-way interaction including multi-attributes. Australian Journal of Agricultural Research, 2006, 57, 1185–1193. https://doi.org/10.1071/AR06081

Ferreira, D. F., Demétrio, C. G. B., Manly, B. F. J., de Almeida Machado, A., & Vencovsky, R. Statistical models in agriculture: biometrical methods for evaluating phenotypic stability in plant breeding. Cerne. 2006, 12(4), 373-388.

Comment 2: 

Here are some minor remarks:

Line 30-32 - the conclusion is true on the basis of the AMMI analysis performed

Line 204-207 - the pattern is so obvious, that it should be omitted

[Response] We have revised the suggestions accordingly.

Comment 3: Table 4 - it would be more interesting to determine the significance of the sowing time effect and not such a combined environmental effect (I suspect that a large impact of the trait variability is caused by nesting sowing time in environmental effect).

Line 447 - I am curious if there is a compatibility in the ranking of genotype between the study traits, as well as between the two sowing dates for individual trait. 

[Response]: As we used a 3-way AMMI, we are not able to revise Table 4, as it shows the combined analyses of ANOVA and PCA for all selected traits. On the other hand, we do agree that the sowing date majorly influenced the experiment, so we added a new paragraph in 3.4., lines 472–476, and added a Table S8 in the supplementary materials, which shows the ranking of all 20 accessions for every trait for the different sowing dates. A comparison of the rankings in Table S8 and Table 6 shows that the rankings are not so different (HFB is the only exception), as the ranking in Table 6 is based on trait means and the stability of the accessions over different environments.

We would like to thank you for pointing out the weaknesses of this study and for your in-depth analysis of our manuscript, as your expert comments helped us to improve our paper and educated us on other statistical methods which we will closely study before submitting the next multifactorial studies of this kind. We hope that our revisions and the herein provided justification for the approach we have taken, are to your satisfaction.

Round 2

Reviewer 2 Report

The authors have explained exhaustively why they could not apply the approach suggested by me. I accept the current form of the manuscript and have no further comments. 

Author Response

We would like to thank Reviewer 2 for his expert comments and suggestions, and we appreciate your time and effort because your comments helped us improve our manuscript.